# Environment-induced heritable variations are common in *Arabidopsis thaliana*

Xiaohe Lin [1,5], Junjie Yin [1,5], Yifan Wang [1,5], Jing Yao[1], Qingshun Q. Li [1,2], Vit Latzel[3], Oliver Bossdorf [4] & Yuan-Ye Zhang [1] ✉

Parental or ancestral environments can induce heritable phenotypic changes, but whether such environment-induced heritable changes are a common phenomenon remains unexplored. Here, we subject 14 genotypes of *Arabidopsis thaliana* to 10 different environmental treatments and observe phenotypic and genome-wide gene expression changes over four successive generations. We find that all treatments caused heritable phenotypic and gene expression changes, with a substantial proportion stably transmitted over all observed generations. Intriguingly, the susceptibility of a genotype to environmental inductions could be predicted based on the transposon abundance in the genome. Our study thus challenges the classic view that the environment only participates in the selection of heritable variation and suggests that the environment can play a significant role in generating of heritable variations.

Heritable variation is the foundation of biodiversity and adaptation to the changing climate[1,2], ultimately generated through mutations and moulded by forces of environmental selection—a tenet well established in textbooks[3]. Several pioneering studies suggest that heritable variation is also directly introduced by parental or ancestral environments[4–7], causing stably transmitted phenotypic and gene expression changes that potentially enhance offspring resistance to environmental stress[4,7,8]. However, whether such environment-induced heritable changes are a common phenomenon remains unexplored, and it is also unknown whether the phenomenon is an occasional exception or a fundamental principle. Therefore, exploring the prevalence of environment-induced heritable changes will challenge the textbook knowledge of the origin of heritable variation and provide critical insights for predicting the adaptive response to global environmental changes.

The phenomenon in which the parental or ancestral environment induces offspring phenotypic changes is conventionally referred to as the parental or transgenerational effect[4], and the major challenge in understanding the prevalence of this phenomenon could be the scale of the experiments[9]. Expanding the scale of these experiments to include different genetic backgrounds has revealed that such environment-induced changes are genotype-specific[10,11] and predictable based on the climate of the genotype's origin[12,13], while this predictability decreases when sacrificing the number of genotypes to include various environmental treatments[9]. Nevertheless, these studies span only one or two offspring generations, and it is unclear whether environment-induced changes are heritably stable. A large-scale experiment encompassing multiple genotypes, environments, and generations is necessary to reveal the prevalence and predictability of environment-induced heritable changes.

In this work, to address the prevalence and predictability of environment-induced heritable changes, we subject 14 natural accessions (genotypes) of *Arabidopsis thaliana* to the Control and ten environmental treatments to establish the ancestral generation. Before this generation, we plant the seeds of these accessions in the control environment for two generations to remove potential influence due to seed collection or storage. Then, we cultivate the offspring of the ancestral generation in the control environment to assess whether those environmental treatments induce phenotypic changes in offspring and whether the induced changes are heritable over four offspring generations (Test I, Fig. 1, Supplementary Fig. 1a, Supplementary Table 1). To explore the reproducibility of

[1]Key Laboratory of the Ministry of Education for Coastal and Wetland Ecosystems, College of the Environment and Ecology, Xiamen University, Xiamen, Fujian, China. [2]Biomedical Sciences, College of Dental Medicine, Western University of Health Sciences, Pomona, CA, USA. [3]Institute of Botany of the CAS, Zamek 1, 252 43 Pruhonice, Czech Republic. [4]Institute of Evolution & Ecology, University of Tübingen, Auf der Morgenstelle 5, 72076 Tübingen, Germany. [5]These authors contributed equally: Xiaohe Lin, Junjie Yin, Yifan Wang. ✉e-mail: zhangyuanye@xmu.edu.cn

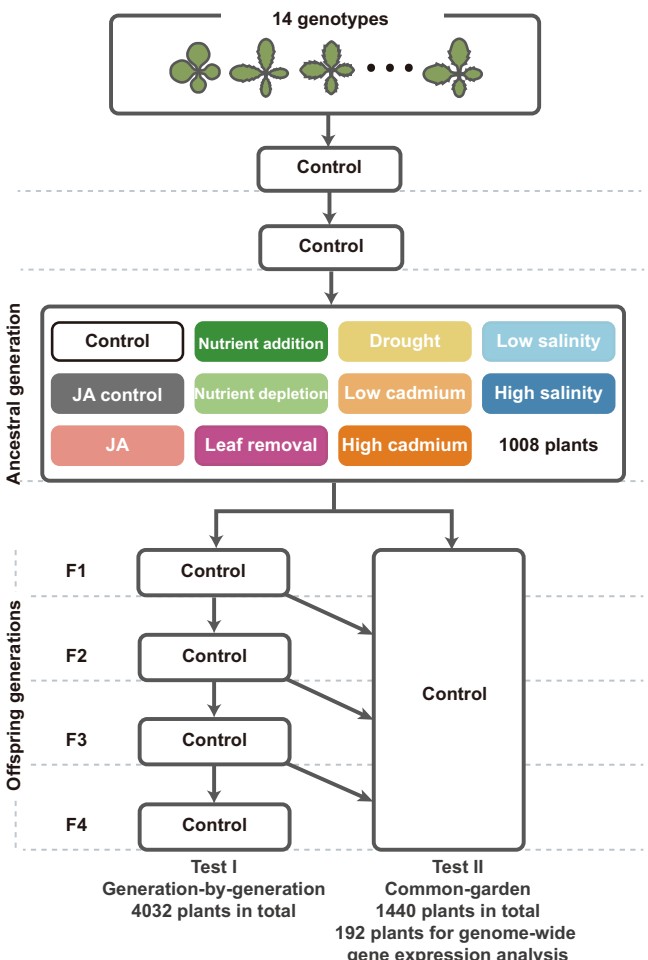

**Fig. 1 | Design of the experiment.** The experiment subjected 14 genotypes of *Arabidopsis thaliana* to one control and ten environmental treatments and lasted seven generations. Plants were grown for two generations before launching the "ancestral generation" to eliminate variation due to seed collection or storage. The ten ancestral environmental treatments included jasmonic acid addition (JA), control for JA, nutrient addition, nutrient depletion, leaf removal, drought, low cadmium, high cadmium, low salinity and high salinity. The control and each treatment group consisted of 12 and six replicates, resulting in a total of 1008 lines. To test the ancestral environmental effect on offspring, we established Test I to grow all these lines in the control environment over four successive generations (F1 to F4), following the single-seed descendant approach. To explore the reproducibility and assess the gene expression changes, we established Test II, where we selected ten genotypes and six environments and planted all four offspring generations together in the control environment.

environment-induced changes, we plant all four offspring generations together under uniform conditions to eliminate any heterogeneity between generations in the growth conditions (Test II, Fig. 1). Besides performing phenotypic observations, we analyse the genome-wide gene expression of samples from Test II to investigate whether heritable phenotypic changes are predicted by heritable gene expression changes and whether the induced expression changes were nonrandom with respect to environments. We find that all ancestral environmental treatments caused heritable phenotypic and gene expression changes. The induced phenotypic changes are quantitatively reproducible between offspring generations and tests. The susceptibility of a genotype to environmental inductions could be predicted based on the transposon abundance in the genome. Therefore, our study reveals that environment-induced heritable changes are common, reproducible, and predictable, and highlights the significance of the environment in generating heritable variations.

## Results

### Prevalence of environment-induced heritable phenotypic changes

In the ancestral generation, we subjected *A. thaliana* genotypes to the control environment and 10 treatment environments: jasmonic acid addition (JA), control for JA, nutrient addition, nutrient depletion, leaf removal, drought, low and high concentrations of cadmium, and low and high concentrations of salinity. The statistics showed that the genotype, environmental treatment and their interaction significantly affected all phenotypes measured (except for the interaction effect on flowering time, Supplementary Table 2). Among the specific treatments, nutrient addition significantly increased aboveground biomass, and jasmonic acid addition slightly enhanced fruit production, representing benign environments (Supplementary Fig. 2a). Other treatments significantly decreased fruit production and thus represented stressful environments (Supplementary Fig. 2a). High concentrations of cadmium and salinity generated more stressful conditions than the corresponding low-concentration treatments, resulting in either smaller plants with fewer fruits or shorter plants with fewer fruits (Supplementary Fig. 2a). Therefore, our experiment provided a gradient of benign and stressful environments to assess their effects on offspring phenotypes.

In the offspring generations, we found little evidence of significant main effects of the ancestral environments on phenotypes but significant genotype-by-ancestral_environment interactions (G × A.E) on all phenotypes measured except for fruit number (Supplementary Fig. 2b, Supplementary Table 2). More importantly, the G × A.E interactions remained significant in the four successive offspring generations. These results demonstrated that the effects of the ancestral environments on offspring phenotypes (i.e., transgenerational effects) depended on the genotypes and persisted for four offspring generations. To understand the stable transmission of such genotype-specific changes caused by ancestral environments, we estimated the effect sizes of changes for different genotypes and environments and plotted them in the order of generations (Fig. 2a). The plot showed considerable variation in the magnitude and direction of the phenotypic changes. In some notable cases, such as the response of genotype Abd-0 to high cadmium and the response of TRE-1 to high salinity, we found that the environment-induced phenotypic changes were stably transmitted over four generations of offspring (Fig. 2a). Therefore, our study demonstrates that environment-induced phenotypic changes are genotype-specific and can be stably transmitted (i.e., are heritable) for at least four offspring generations.

While several cases of environment-induced heritable changes have been identified, a fundamental question remains: how prevalent are these changes across different environments? To answer this question, we transformed the quantitative estimates of effect sizes into qualitative assessments, defining an environment-induced change as occurring when the effect size significantly deviated from zero. The occurrence data revealed environment-induced phenotypic changes in each genotype and environment (Fig. 2b). Importantly, a substantial proportion (36.8% 10.4%, and 7.5%) of these changes occurred in two, three, and four offspring generations, suggesting that these environment-induced changes were heritable. Furthermore, we observed heritable changes (changes occurring in at least two generations) in all environments and persistent changes (changes occurring in three generations and remaining significant in the fourth generation) in seven environments (Fig. 2b). Consequently, these findings demonstrate that environment-induced heritable changes commonly occur across various environments.

### Predictability and reproducibility of environment-induced phenotypic changes

The qualitative data revealed that the genotype, ancestral treatment, offspring generation, and phenotype significantly affected the

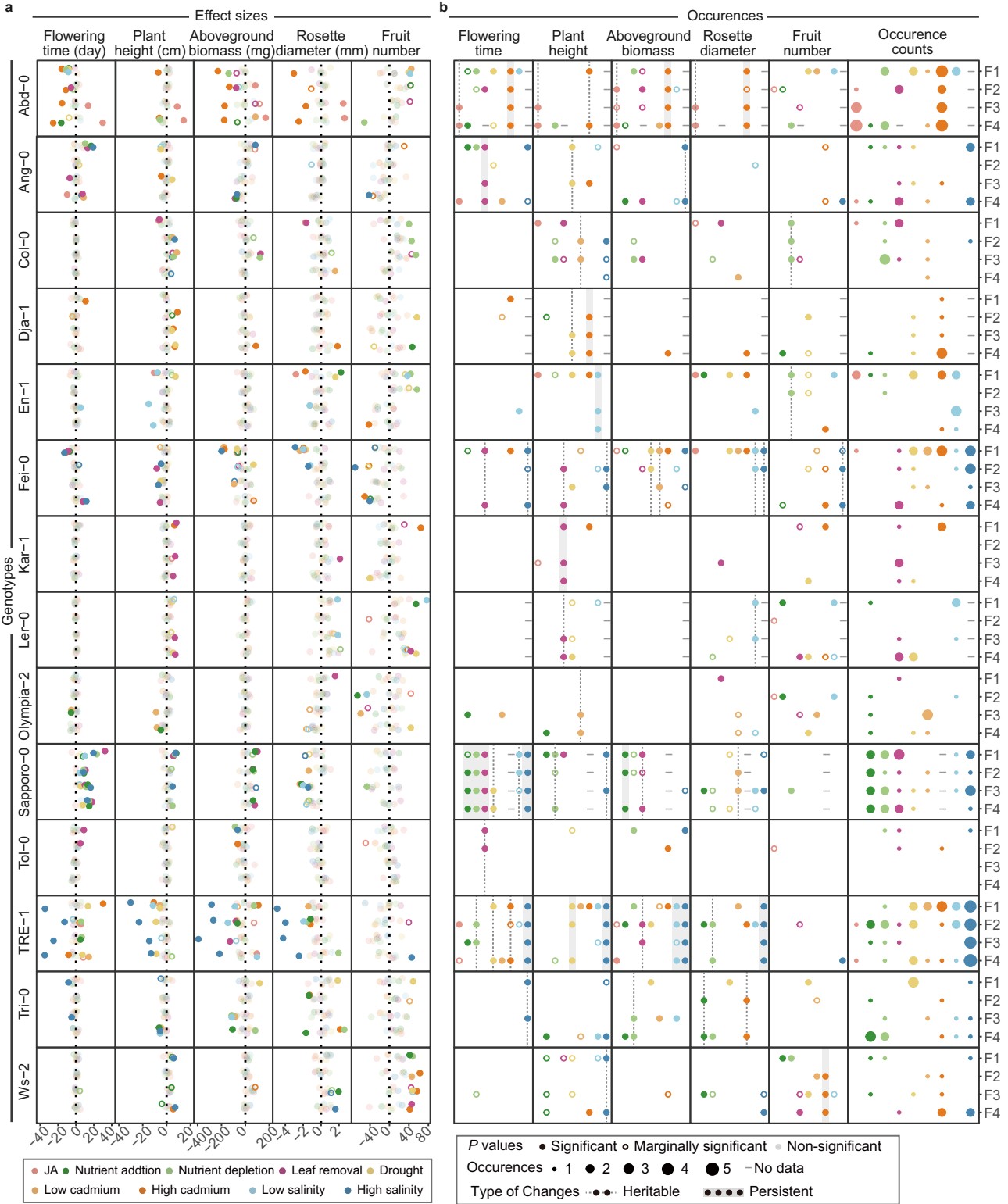

**Fig. 2 | Prevalence of environment-induced heritable phenotypic changes.**
**a** The effect sizes of environment-induced phenotypic changes measured in Test I. The effect sizes are extracted from linear mixed-effects models (LMM) and two-sided tests are applied. **b** The occurrence of significant and marginally significant occurrence of environment-induced changes and this occurrence summed over phenotypes. Environment-induced heritable changes (significant changes in at least two generations) marked with dash lines are found in each environment. Source data including the statistics and exact $P$ values are provided as a Source Data file.

occurrence of environment-induced phenotypic changes (Fig. 3a). The occurrence frequency estimated on the fitness proxy ( = number of siliques per plant or fruit number) was the lowest among different phenotypes, marginally significantly lower than those of flowering time and plant height. This frequency also varied across different

offspring generations, but no consistent decline was observed over successive generations. In general, the magnitude of the ancestral treatment effect did not predict the frequency of occurrence. For instance, the leaf removal treatment, affecting only fruit number, appeared to be much less stressful than the low-salinity treatment

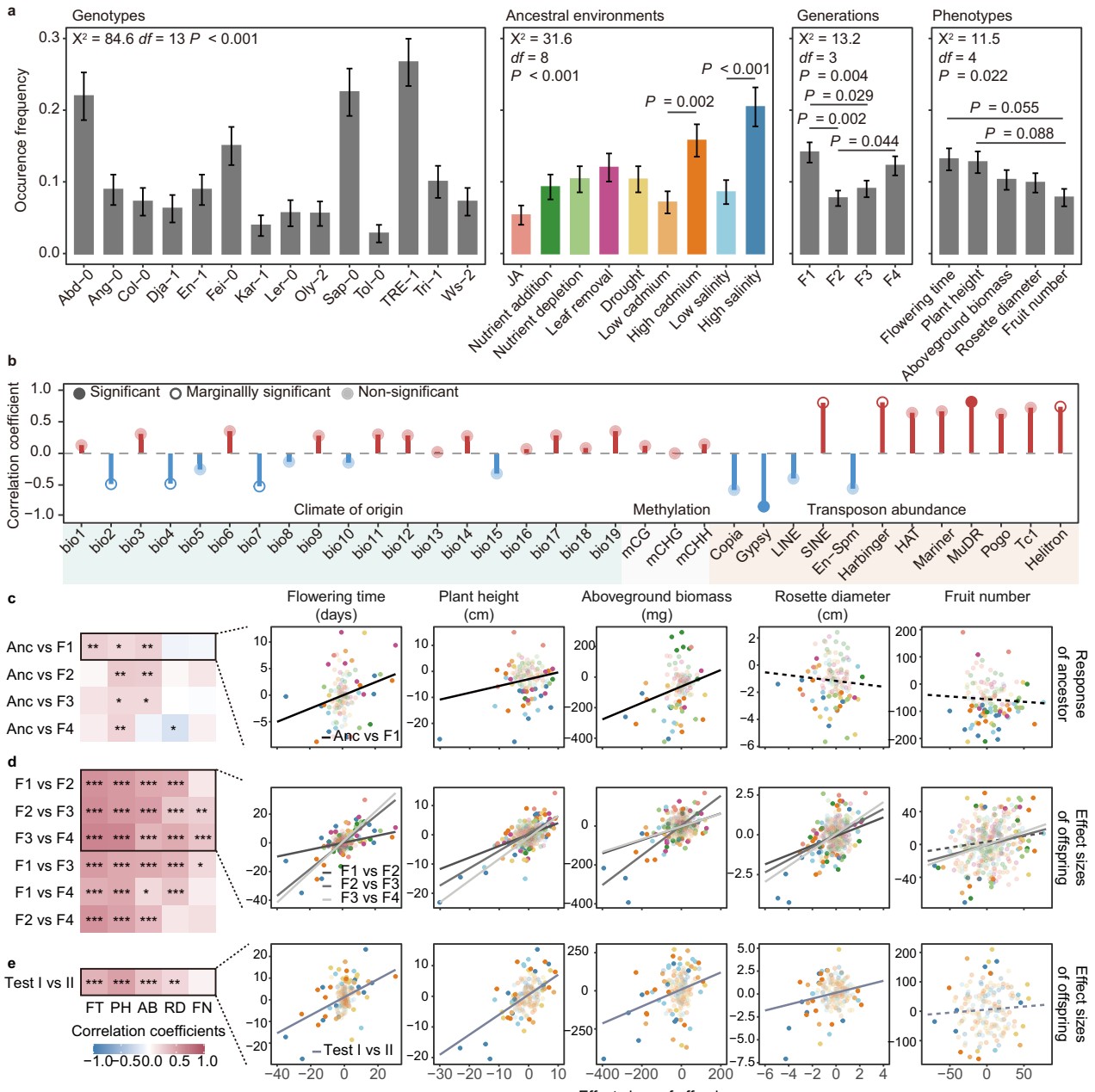

**Fig. 3 | Predictability and reproducibility of environment-induced phenotypic changes. a** The occurrence frequency (mean ± SE) of significant environment-induced changes varied between genotypes, ancestral environmental treatments, offspring generations and phenotypes. The occurrence frequencies are estimated from six independent biological replicates ($n = 6$) per genotype/treatment/generation. Significance is tested using generalized linear mixed-effects models (GLMM) with two-sided chi-square statistics. The $P$ values for multiple comparisons are adjusted with the "single-step" method in the function *glht* (R package "multcomp"). **b** The occurrence frequency of different genotypes could not be predicted from the climate at origin or methylation, but from the abundance of transposons in certain superfamilies. **c** The effect sizes of induced changes show limited correlations with ancestor response. **d** The effect sizes are quantitatively reproducible between offspring generations. **e** The effect sizes are quantitatively reproducible between Test I and Test II. Solid circles represent significant effect sizes in either the generation or test, and transparent circles represent nonsignificant effect sizes in both generations and tests. FT Flowering time, PH Plant height, AB Aboveground biomass, RD Rosette diameter, and FN Fruit number. Source data are provided as a Source Data file.

(affecting four phenotypes), but the occurrence frequencies associated with these two treatments were similar (Supplementary Fig. 2a, Fig. 3a). Only within the same treatment type was a higher intensity of treatment associated with an increase in the occurrence frequency of induced phenotypic changes (high vs. low concentrations of cadmium/salinity, Fig. 3a).

Among the factors affecting the occurrence of environment-induced changes, genotype emerged as the primary explanatory variable. This factor accounted for the largest proportion of the variance in occurrence frequency (Fig. 3a), and removing the three genotypes with the highest frequencies (TRE-1, Abd-0, Sap-0) rendered the effect of ancestral environments on occurrence insignificant (Supplementary Fig. 3a). To explore which factor predicted the susceptibility of a genotype to environmental induction, we regressed the occurrence frequency of genotypes against different explanatory variables (climate of origin, methylation levels, and transposon abundance) and found that the abundance of transposons in the Gypsy ($r = -0.840$) and MuDR ($r = 0.815$) superfamilies explained a significant

proportion of the variation (Fig. 3b, Supplementary Fig. 3b, Supplementary Table 1). Therefore, these findings suggest that genotype may be a primary predictor of the occurrence of environment-induced changes, and this property of a genotype can be predicted by the number of transposons in its genome.

Although the occurrence of environment-induced changes is predictable to some extent, predicting the effect sizes of these changes proved challenging. Consistent with a previous study[9], the responses of the ancestors showed little, if any, predictability related to the effect sizes of offspring (Fig. 3c). Nevertheless, it was encouraging to find that the effect sizes of offspring could exceed the range of ancestral responses, as observed for flowering time (Fig. 3c). For this trait, the fixed G × A.E effect also explained a greater proportion of the total variation than G × E (Supplementary Fig. 3c). Compared to ancestor-offspring correlations, the coefficients estimated between offspring generations were significantly greater and consistently significantly deviated from zero, except for fruit number, for which the coefficient reached 0.682 (median = 0.474) (Fig. 3d, Supplementary Fig. 4a, b). The coefficient increased to 0.857 (median = 0.614) when including only significant effect sizes in the analysis and declined substantially and insignificantly deviated from zero for most cases when excluding significant effect sizes from the analysis (Supplementary Fig. 4c). A similar pattern of correlation coefficients was also observed when comparing effect sizes between Test I and Test II (Fig. 3e, Supplementary Fig. 4a, please also see below for Test II). Intriguingly, when we scaled down the experiment to include only one genotype (but multiple treatments) or one treatment (but multiple genotypes), we found that the chances of obtaining significantly correlated effect size correlations between generations declined substantially (Supplementary Fig. 4d). Therefore, our study demonstrates that the effect sizes of environment-induced phenotypic changes are reproducible between offspring generations and tests, and conducting large-scale experiments to obtain abundant significant effect sizes is crucial for achieving this reproducibility.

Despite the findings regarding the predictability and reproducibility of environment-induced changes, some unresolved questions remained: why did some significant phenotypic changes disappear in one generation but reappear in the next (Fig. 2b), and why did the occurrence frequency of induced changes vary between offspring generations (Fig. 3a)? A potential explanation is that the growth conditions differed between offspring generations, as evidenced by significant phenotypic variation between generations in the control group (Supplementary Fig. 2c, Supplementary Table 3). Such variation may also contribute to the inconsistent effect sizes observed between generations (significant G × A.E × F, Supplementary Table 4). To eliminate the influence of variable growth conditions, we conducted a second test of ancestral treatment effects (Test II), where we selected a subset of 10 genotypes and five treatments and grew F1 to F4 plants together under the same conditions (Fig. 1). Consistent with the results of Test I, Test II revealed that the effect of ancestral environments depended on the genotypes (Supplementary Fig. 5a, Supplementary Table 5); that environment-induced heritable changes were common across environments (Supplementary Fig. 6); that the occurrence frequency of these changes varied significantly between genotypes (Supplementary Fig. 7a); and that the variation among genotypes could be predicted by the abundance of several transposon superfamilies, including HAT ($r = 0.893$, $P = 0.041$), Mariner ($r = 0.873$, $P = 0.053$), and MuDR ($r = 0.846$, $P = 0.071$) (Supplementary Fig. 7b, c).

Surprisingly, the repeatability of the effect sizes between generations in Test II did not increase compared to that in Test I (Fig. 3d, Supplementary Fig. 7d). We still found significant phenotypic variations between generations in the control group (Supplementary Fig. 5b, Supplementary Table 6), and the genotype-specific ancestral treatment effects varied significantly between generations for flowering time and aboveground biomass (significant G × A.E × F,

Supplementary Table 7). As all plants in Test II were grown under the same conditions, the observed generation effects could only be introduced by different growth conditions of previous generations in Test I. Taken together, these results suggest that both the current conditions and conditions of previous generations can affect the measurements of environment-induced changes, and conducting large-scale experiments under well-controlled conditions is the key to achieving high reproducibility and an in-depth understanding of environment-induced changes.

## Environment-induced heritable gene expression changes

In Test II, we conducted transcriptomic analysis of offspring generations (F1-F4) to investigate genome-wide gene expression changes induced by ancestral environments (Fig. 1). The patterns of genome-wide gene expression were primarily determined by genotypes (Supplementary Fig. 8a), with noticeable treatment-specific differentiation observed only in the high-cadmium treatment of Abd-0 and drought treatment of TRE-1 (Supplementary Fig. 8b). The number of differentially expressed genes (DEGs) induced by the ancestral treatment ranged from 146 to 3694 (Fig. 4a), which varied significantly between genotypes but not between environmental treatments or generations (Fig. 4b). Importantly, a significant proportion (15.5%, 3.43%, and 1.54%) of these genes exhibited differential expression in two, three, and four offspring generations (Fig. 4c), indicating that the gene expression changes were heritable. We considered those genes that were differentially expressed for at least two generations to be heritable DEGs, and the number of heritable DEGs ranged from 69 to 2668. Importantly, we identified heritable DEGs in each genotype and environment; heritable DEGs that remained differentially expressed over four offspring generations were also found in each environment (Fig. 4c). The number of heritable DEGs for each genotype and treatment was significantly related to the occurrence frequency of induced phenotypic changes (Supplementary Fig. 9a) and was also significantly explained by the number of transposons in several superfamilies (Fig. 4d, Supplementary Fig. 9b). Therefore, these results demonstrate that environment-induced heritable gene expression changes were common across different environments, and the number of genes showing heritable expression changes correlates with the abundance of transposons in the genomes.

An important unanswered question is whether the induced gene expression changes are directional or unidirectional. If these changes are directional, we expect the DEGs induced by different environments to have distinct functions. To test this hypothesis, we conducted GO and KEGG functional enrichment analyses of heritable DEGs and found that 80.2% (73 out of 91) of enriched GO terms and 58% (27 out of 46) of enriched KEGG pathways were shared between treatments (Fig. 5a, b, Supplementary Fig. 9c, d). The overlapping functions appeared to be primarily associated with the stress response (Fig. 5a) and metabolic pathways (Supplementary Fig. 9c). These results indicated that, contrary to our expectations, heritable DEGs induced by different environmental treatments were functionally similar, which may be associated with these treatments (drought and low and high concentrations of cadmium) generating similar stresses (i.e., inducing hypoxia in the root system).

Intriguingly, in addition to being shared between different environmental treatments, a substantial proportion of terms/pathways were also shared between genotypes (Fig. 5b, Supplementary Fig. 9d). The shared terms/pathways likely originated from heritable DEGs that were shared between treatments and/or genotypes (Fig. 5b, c, Supplementary Fig. 9d, e). As the heritable expression changes in these genes were repeatedly induced in two or more treatments or genotypes, these genes could be considered susceptible to environmental induction. We hypothesize that the susceptibility of these genes is explained by certain genomic features, such as the abundance of transposons in their promoter and gene body regions. To test this

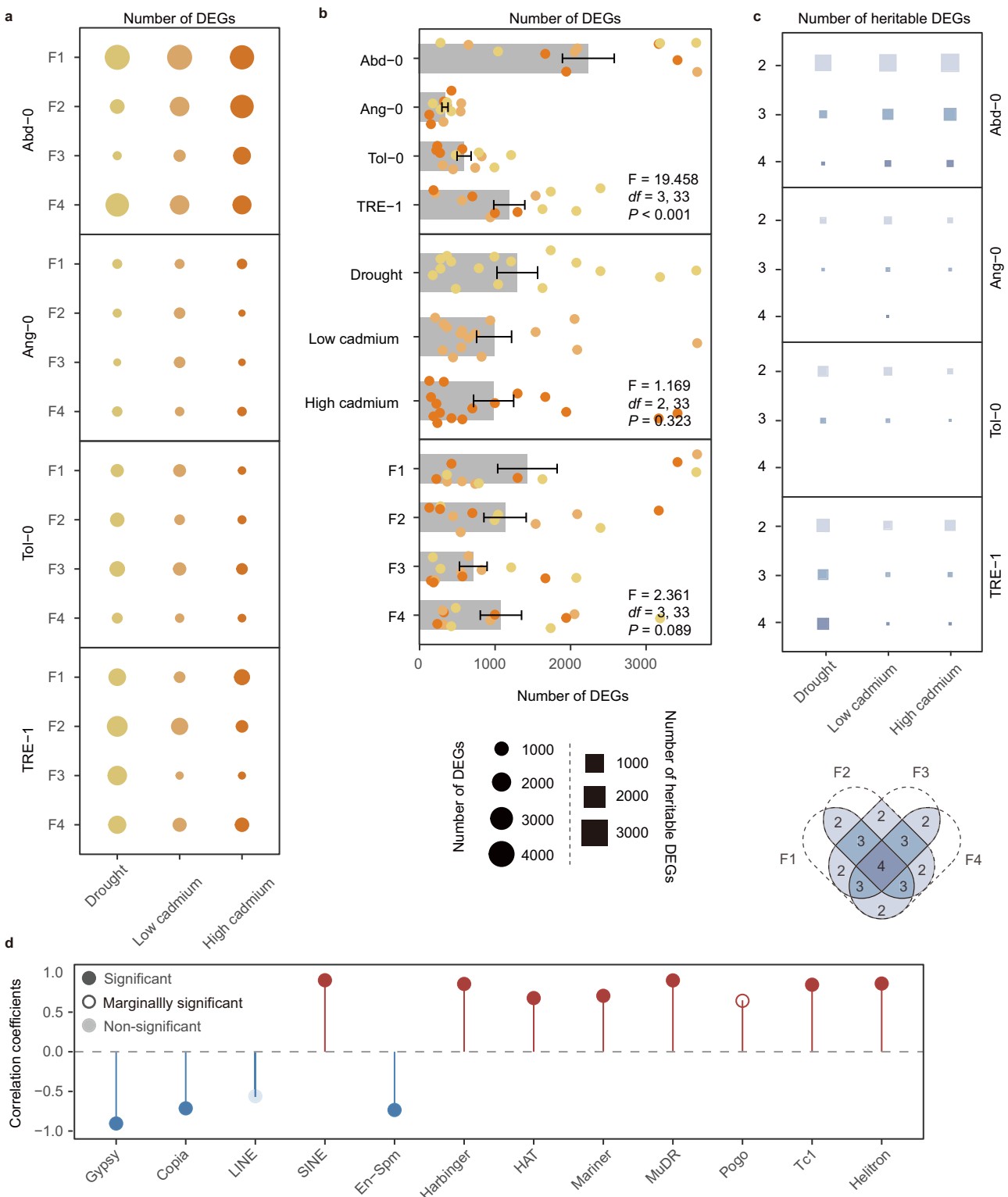

**Fig. 4 | Environment-induced heritable gene expression changes are found in each ancestral environmental treatment, and the predictability of environment-induced gene expression changes. a** The number of differentially expressed genes (DEGs) for each genotype, ancestral environment treatment and offspring generation. **b** The number of DEGs (mean ± SE) varied significantly between different genotypes, but not between treatments and generations. The number of DEGs are estimated from three independent biological replicates (*n* = 3)

per genotype/treatment/generation. Significance is tested using linear models with two-sided *F* statistics. **c** The number of heritable DEGs (genes significantly differentially expressed in two, three or four offspring generations) for each genotype and treatment. **d** The number of heritable DEGs for different genotypes could be predicted from the abundance of transposons in several superfamilies. Source data are provided as a Source Data file.

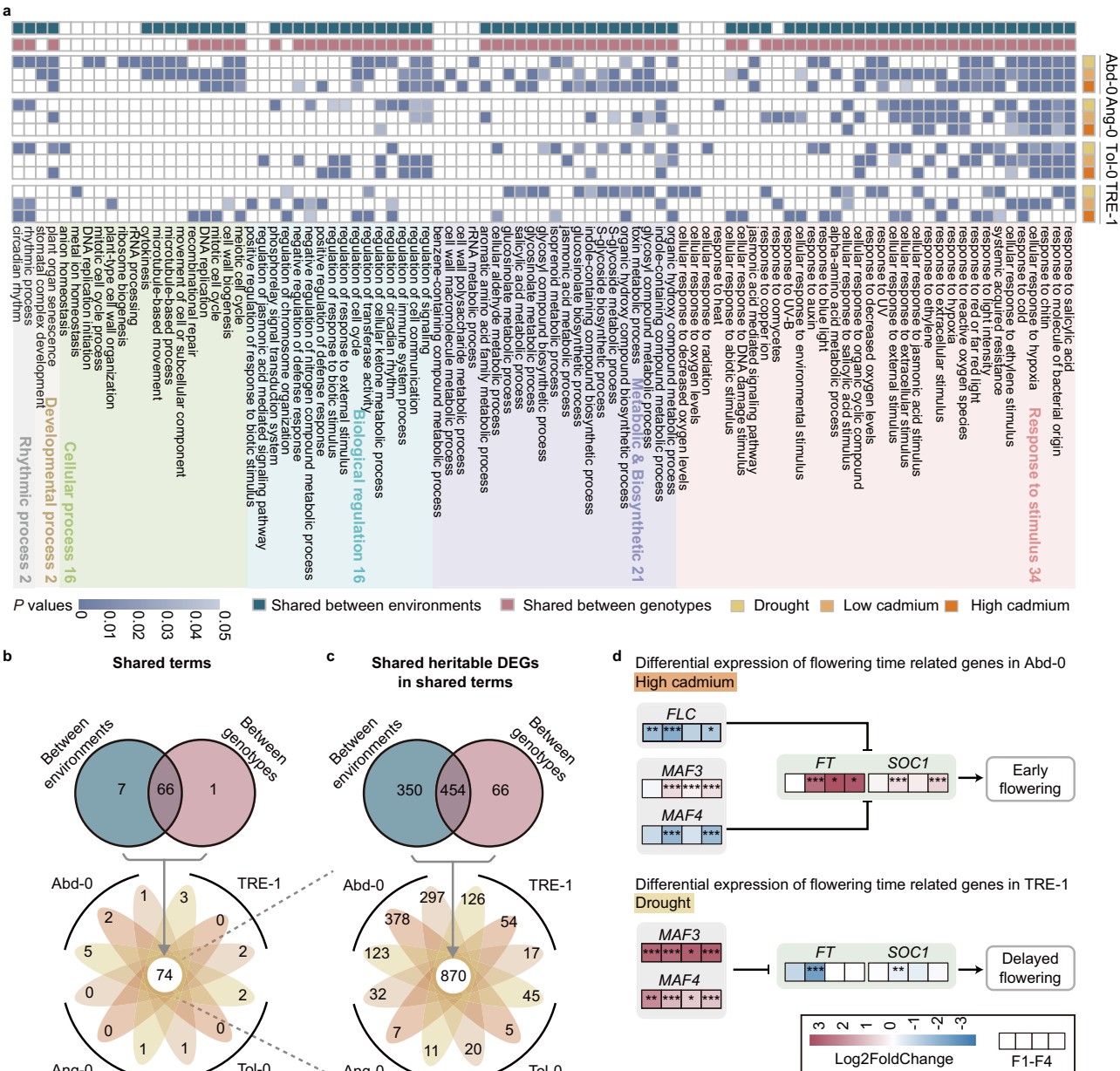

**Fig. 5 | Heritable gene expression changes induced by various environments were functionally similar and explained the induced phenotype changes.** **a** Gene ontology (GO) enrichment results of heritable differentially expressed genes (DEGs) show that induced gene expression changes are functionally similar between environmental treatments and genotypes. Significance was calculated by hypergeometric distribution by R package clusterProfiler. **b** Venn plot shows the numbers of shared terms between environmental treatments and genotypes. **c** Venn plot shows the number of heritable DEGs within the shared terms. **d** The heritable DEGs explained the environment-induced heritable changes in flowering time. Source data are provided as a Source Data file.

hypothesis, we identified all heritable DEGs shared between genotypes or treatments (Supplementary Fig. 10a) and found that the abundance of members of several superfamilies, including Gypsy, HAT, Mariner, and Helitron, upstream and body of these susceptible genes was significantly higher than that of background genes (Supplementary Fig. 10b). Therefore, these results suggest that environment-induced changes may be nonrandom in the sense of genomic features.

To investigate whether the heritable phenotypic changes were associated with heritable gene expression changes, we focused on the phenotype of flowering time, which has a comparatively well-characterized molecular basis[14–16]. Among the 79 genes underlying flowering time changes identified in the literature, 16 were differentially expressed in at least one genotype or treatment, and seven were heritable DEGs (Supplementary Fig. 10c). The heritable gene

expression changes of six genes co-occurred with heritable flowering time changes, including the early flowering of Abd-0 in response to high cadmium and drought-induced late flowering in TRE-1 in response to drought (Fig. 5d, Supplementary Fig. 6). In Abd-0, ancestral treatment with high cadmium caused downregulation of *FLC* and *MAF4* and upregulation of *FT* and *SOC1*, promoting early flowering (Fig. 5d). In TRE-1, ancestral drought treatment caused the upregulation of *MAF3* and *MAF4* and downregulation of *FT* and *SOC1*, which was associated with delayed flowering time (Fig. 5d). *FLC* is the primary determinant of the vernalization requirement and negatively regulates the expression of *FT*[17,18]. *MAFs* are paralogues of *FLC*, which act redundantly with *FLC* as floral repressors[17,19]. Therefore, these results suggest that heritable expression changes in key genes underlie environmentally induced heritable phenotypic changes.

## Discussion

Here, we conducted a large-scale multigenerational experiment, and the main finding of our study was that environment-induced heritable changes were caused by each ancestral environmental treatment. These results extend the understanding of the phenomenon of environment-induced heritable changes beyond occasional findings under specific environmental conditions[4,5,12] and suggest that this phenomenon could be a fundamental principle of genetics. Our work challenges the textbook understanding that environments are only involved in the selection of heritable variation[3] and suggests that environments are broadly involved in the generation of heritable variation.

The scale of our experiment allowed us to explore the stochasticity of environment-induced heritable changes and reveal genotypes with a higher abundance of transposons that are more susceptible to environmental inductions. Furthermore, our study revealed that genes with more transposons upstream are also prone to environmental induction. These results indicate that transposons play a central role in mediating environment-induced changes, which could occur through epigenetic modifications of transposons (i.e., DNA methylation[20,21] and RNA interference[22]) that regulate the expression of downstream genes or through de novo transposon insertions[23,24]. Transposons are mobile elements that constitute a significant proportion of eukaryotic genomes[25]. While these elements were once considered junk DNA, it became evident that they can contribute to the transcriptional binding region[26], methylation loci[21], transcriptional isoform sites[27], and even coding sequences[28] to the host genomes. Barbara McClintock, who discovered transposons, proposed the idea of "genome shock," suggesting that stress can activate the movement of transposons, causing insertional mutations and structural variants[29]. This viewpoint has been increasingly supported by evidence that stress increases transposition rate[30–33], with the phenotypic effects of de novo transposon insertions including enhanced resistance of insects to pesticides[34–36] and decreased sensitivity of plants to abiotic stress[36–39]. The stress-induced transposition may be mediated by Hsp70 chaperone, a response factor to various biotic and abiotic stressors in plants and animals[40,41], whose inducible expression could disturb the biogenesis of transposon-silencing piRNA[42]. These studies, along with our own, bridge and revive these previous ideas and explanations, suggesting that the environment is not only the selector but can also act as an inducer of genetic variation[36,43].

The finding that environment-induced heritable changes are common across environments also indicates that, besides standing genetic variation within populations[44,45], de novo heritable variations caused by environmental changes can serve as another primary source for rapid adaptation. These de novo heritable variations may arise from the mechanisms discussed earlier, including epigenetic changes (DNA methylation[46], histone modifications[47], and small interfering RNAs[6]) or de novo transposon insertion[23,24,48]. Some notable examples of rapid evolution based on de novo transposon insertion include the industrial melanism mutation of the peppered moth[23] and the rapid increase in resistance to DDT in *Drosophila melanogaster*[24]. While many transposon insertions induced by environmental factors are deleterious, they enable the creation of alleles with significant, positively selected effects[37,49,50]. If the environment can act as both a selector and an inducer of heritable variation, then the evolutionary speed constraints imposed by standing genetic variations will be liberated[43,48]. Therefore, exploring environmentally induced heritable variations and the molecular mechanisms related to transposon regulation is not only significant for fundamental research in genetics and evolution, but also crucial for assessing the threats of global changes and species responses.

## Methods

### Plant materials and the experimental design

*Arabidopsis thaliana* is a model plant species for molecular biology research and is increasingly used to address questions in ecology and evolution[51]. It is a self-pollinating annual weed that originates from North Africa[52] and is broadly distributed across Europe, Asia, and North America. Seeds collected from different geographic sites, known as natural accessions or ecotypes, are preserved at the *Arabidopsis* Biological Resource Centre (ABRC). As this species is predominantly selfing, different accessions are also taken as different genotypes[10,12]. Here, we adopted 14 *A. thaliana* genotypes ordered from the ABRC, including Abd-0 (from Aberdeen, UK), Ang-0 (Namur, Belgium), Col-0 (Columbia, US), Dja-1 (Chui, Kyrgyzstan), En-1 (Frankfurt, Germany), Fei-0 (Aveiro, Portugal), Kar-1 (Suusamyr, Kyrgyzstan), Ler-0 (Munich, Germany), Olympia-2 (Olympia, Greece), Sap-0 (Hokkaido, Japan), Tol-0 (Ohio, USA), TRE-1 (Marne, France), Tri-0 (Sevilla, Spain) and Ws-2 (Belarus, Belarus) (Supplementary Fig. 1a, Supplementary Table 1). Four seeds per genotype were planted in a common environment for two generations to eliminate the variation due to seed collection or storage. In the second generation, seeds of the four resulting plants were pooled for each genotype to establish the ancestral generation.

To investigate whether environmental changes caused heritable changes in offspring (i.e., transgenerational effects), we subjected the ancestral generation to different environmental treatments and assessed the offspring phenotypic changes over four generations in a common environment. The ancestral environments included the control and ten treatments: jasmonic acid addition (JA), control for JA, nutrient addition, nutrient depletion, leaf removal, drought, low cadmium, high cadmium, low salinity and high salinity (Fig. 1). Each treatment included six replicate plants per genotype, and the control included 12 plants per genotype, resulting in 1008 plants grown or lines established in the ancestral generation. Seeds were collected and reserved for each line, and in the offspring generation, we planted one descendant per line, following the single-seed descendant approach. In Test I, all 1008 lines were planted generation-by-generation in the control environment for four generations (F1-F4), and together 4032 plants were planted. For each offspring generation, seeds were collected and reserved for each line and generation separately.

The growth room contained four shelves, each with four layers (Supplementary Fig. 1b). In each layer, we organized the plants into six rows and 14 columns, and this organization allowed one replicate per treatment/genotype to be planted in two layers. For one replicate of the ancestral generation, treatments were randomly assigned to rows in two layers, and genotypes were distributed randomly among columns in a row (Supplementary Fig. 1c). For the offspring of this replicate, we assigned them to two random layers and distributed the plants randomly among different positions (Supplementary Fig. 1c). All randomization was conducted using the "*RAND*" function in Excel.

To eliminate the potential influence of growth conditions that varied between generations, we planted all four offspring generations together in the control environment (Test II). Due to limited space in the growth room, Test II included a subset of ten genotypes (Abd-0, Col-0, Ang-0, Dja-1, Fei-0, Ler-0, Sapporo-0, Tol-0, TRE-1 and Tri-0), six different environments (the control, drought, low cadmium, high cadmium, low salinity and high salinity), and thus 360 lines (=10 genotypes * 6 environments * 6 replicates). For each line, we regenerated one plant from reserved seeds of the previous generation, and 1440 plants were grown to establish F1-F4. We adopted a completely random design, assigning the 1440 plants randomly into positions among the 16 layers, with six rows and 15 columns each (Supplementary Fig. 1c). Due to the insufficient storage of seeds, especially for the F1 generation, Test II eventually involved 1252 plants. The randomization was conducted using the "*RAND*" function in Excel.

## Growth conditions, phenotype measurements, and sample collection

The experiment was conducted by four graduate students who assisted each other (Supplementary Fig. 1d). Prior to sowing, we placed 10–15 seeds in a 2 mL centrifuge tube, added 1 mL of sterilized distilled water, and kept the tubes in a 4 °C freezer in the dark for 7 days. The soaked seeds were then evenly distributed into pots (5.5 × 5.5 × 6.0 cm) filled with a mixture of 1:1 sterilized soil and vermiculite. The plants were subjected to a photoperiod of 16 h light (7:00 am to 11:00 pm) and 8 h dark (11:00 pm to 7:00 am), with temperatures ranging between 22 and 24 °C. On the tenth day, one *A. thaliana* seedling was randomly selected and retained. Lines that failed to germinate were transplanted by seedling of a randomly selected replicate from the same genotype and treatment. For the control environment, we watered the plants from the bottom every four days and substituted water with half Hoagland solution (Hope Bio-Technology Co., Ltd) every 16 days. For the jasmonic acid addition (JA) treatment, we sprayed the plants with 100 µmol/L jasmonic acid solution (Sigma-Aldrich) dissolved in 0.5% ethanol every four days, and in the control for JA treatment, we sprayed 0.5% ethanol (sinopharm Chemical Reagent Co.,Ltd) solution to plants. For the nutrient addition treatment, we watered plants with half Hoagland solution every four days and never gave Hoagland solution to plants in the nutrient depletion treatment. For the leaf removal treatment, we removed half of the total number of leaves 30 days after germination. For the drought treatment, plants were watered every eight days. We dissolved $CdCl_2$ (Shanghai Macklin Biochemical Technology Co., Ltd.) in water or half Hoagland solution to form 500 µmol/L and 1500 µmol/L $CdCl_2$ solutions and watered the plants with these solutions in low- and high-cadmium treatments. For the low- and high-salinity treatments, we watered the plants with 50 mmol/L NaCl solutions (Sinopharm Chemical Reagent Co.,Ltd.). These treatments started 25 days after sowing and lasted for 32 days. Seeds for the next generation were collected 50–60 days after sowing. In the ancestral generation, 26 out of 1008 (2.58%) lines died before seed production, and their offspring generation was replaced with seeds of a randomly selected replicate from the same genotype and treatment.

We collected phenotypic data from the ancestral and offspring generations in Test I and Test II. Flowering time was checked daily and recorded as the days from sowing to the opening of the first flower. Fruit number was the number of siliques for a plant, used as a proxy for plant fitness. Rosette diameter was measured as the length of the most extended rosette cross centre. Plant height was the height of a plant when pulled straight. Aboveground biomass was the weight of the aboveground parts after drying in an oven at 80 °C for 48 h.

## Statistical analysis of phenotypic data

To assess the effects of the environmental treatment and genotype on the ancestral generation, we applied a linear mixed-effect model to each phenotype (flowering time, plant height, aboveground biomass, rosette leaf diameter and fruit number). The model included genotype (G), environment (E), and their interaction (G × E) as fixed effects and the layer (A.Layer) and row (A.Row) as random effects, which was coded as $Y_{Anc}$ ~ G + E + G:E + (1│A.Layer) + (1│A.Row) (1). Note that the row number was unique and differed between layers.

To assess the effects of ancestral treatment and genotype on offspring generations, we applied a linear mixed-effect model to each phenotype measured in Tests I and II. The model included offspring layer (O. Layer), genotype (G), ancestral environment (A.E), and their interaction (G × A.E) as fixed effects and ancestor layer (A. Layer) and row (A. Row) as random effects and was coded as $Y_{Off}$ ~ O. Layer + G + A.E + G:A.E + (1│A. Layer) + (1│A. Row) (2). We also integrated all four offspring generations in a model. The model included offspring layer (O. Layer), genotype (G), ancestral environment (A.E), and offspring generation (F) and their two and three interactions as fixed

effects and line (Line), ancestral layer (A.Layer) and row (A.Row) as random effects and was coded as $Y_{Off}$ ~ O.Layer + G + A.E + F + G:A.E + G:F + A.E: F + G:A.E:F + (1│Line) + (1│A.Layer) + (1│A.Row) (3). In Test I, generation was nested in the offspring layer, removing the effect of which also removed that of generation, and Model (3) -.- F was applied to Test I. In addition, we applied a generalized linear mixed-effect model to the subsets of data, including only the offspring of the control environment. This model included the genotype (G) and offspring generation (F) and their interaction as fixed effects and was coded as Y ~ G + F + G:F+ (1│A. Layer) + (1│A. Row) (4).

These models were formulated with function *lmer* in the R ver.3.6.2 package "lme4" v.1.1.30[53]. Statistics of fixed effects were estimated with the *Anova* function in the "car" package v3.1.1[54], and those of random effects were estimated with the *ranova* function in the "lmerTest" package v3.1.3[55]. The effect sizes of ancestral responses to environmental treatments for different genotypes were extracted from model (1), and those of induced changes in offspring generations for different genotypes were extracted from model (2) with the *summary* function following Groot et al.[12]. For the JA treatment, the effect sizes were estimated by applying the models to a subset of data consisting of JA and JA control, and for other treatments, the effect sizes were estimated by applying the models to the respective treatment and the control. The significance of effect sizes was obtained with the *summary* function in the "lmerTest" package v3.1.3. According to these significances, we converted the effect sizes to qualitative occurrence data, where significant effect sizes were assigned a value of one, and nonsignificant effect sizes were assigned a value of zero.

To estimate the variance explained by different variables in mixed-effects models (1) and (2), we converted these models into full random effect models and applied the function *VarCorr* (package "lme4") to the models. Model (1) was converted to $Y_{Anc}$ ~ 1 + (1│G) + (1│E) + (1│G × E) + (1│A.Layer) + (1│A.Row), and Model (2) was converted to $Y_{Off}$ ~ 1 + O.Layer + (1│G) + (1│A.E) + (1│G × A.E) + (1│A.Layer) + (1│A.Row). The estimated variance was visualized with the function *geom_scatterpie* in the R package "scatterpie" v0.1.8 [https://github.com/GuangchuangYu/scatterpie].

## Predictability and reproducibility analysis

To estimate whether the genotype, treatment, generation, and phenotype significantly affected the occurrence of transgenerational effects, we applied generalized linear mixed-effect models to the qualitative occurrence data of Test I and Test II. The model included the genotype, treatment, generation and trait as fixed effects and an additional factor group as a random effect. As effect sizes of different traits were estimated from the same group of six replicate plants per genotype/treatment/generation, we defined the six replicates as one group, which was assigned a group ID. The model was formulated with function *glmer* in the R package "lme4" v.1.1.30 and was coded as Y ~ G + A.E + F + Trait + (1│group.id) (5). Statistics of fixed effects were estimated with the *Anova* function in the "car" package. Furthermore, we used the *glht* function in the package "multcomp"[56] v1.4.20 to further investigate the differences in occurrence frequency among different ancestral treatments, generations, and phenotypes. The adjusted P values were calculated with the "single-step" method based on the joint normal or t distribution of the linear function.

To explore which factors could predict the genotypic variation in the occurrence of transgenerational effects, we related the occurrence probability of genotypes to climates at the origin sites, the DNA methylation level, and the number of transposons in the genomes. We extracted climate data (Bio1-Bio19) from WorldClim [https://worldclim.org], including monthly climate data between 1901 and 2000. The data were extracted with the coordinates of genotypes from a 30-s spatial resolution (~1 km²) by the R package "raster" v3.6.20 (Supplementary Table 1). The raw methylation data of 14 genotypes were obtained from published sources[57]. Quality control was

performed using Trimmomatic v0.36[58] with the following parameters: ILLUMINACLIP: TruSeq3-PE.fa:2:30:10" LEADING:20 TRAILING:20 SLIDINGWINDOW:4:20 MINLEN:50. We aligned the reads against the TAIR10 reference with bwa-meth[59] v0.2.5 with the default parameters. The aligned reads with mapping quality higher than 30 were obtained with SAMtools v1.7[60] with the -q 30 option, and duplicates were marked and removed by Picard v2.18.10 [http://broadinstitute.github.io/picard] and SAMtools (-F 1024). The methylation level of each cytosine for different contexts (CG, CHG, and CHH) was extracted with MethylDackel v0.6.1 [https://github.com/dpryan79/MethylDackel]. We excluded cytosines with a read depth lower than 5, and the average methylation level per cytosine was estimated for each context (Supplementary Table 1).

We extracted the number of transposable elements from published sources[61]. The original file contained copy number estimation based on read coverage for the 317 TE families analysed across 211 *A. thaliana* accessions. We summarized the total number of TEs according to superfamilies and then used log-transformed counts for further analysis. These data were available for seven genotypes, including Abd-0, Ang-0, Col-0, Dja-1, Sap-0, Tol-0, and Ws-2 (Supplementary Table 1). The transposon data of Col-0 were not included in the subsequent analysis because Col-0 is the reference genome and shows biased (much more detailed) annotation of transposons compared to other genotypes. We applied arcsine square root transformation to the occurrence frequency and log transformation to the number of transposons in different superfamilies, regressed the transformed occurrence frequency per genotype against the climate data, average methylation levels and transformed transposon number, and extracted the correlation coefficient and *P* value from the regression.

To investigate whether the effect sizes of environment-induced changes can be quantitatively predicted from the ancestor's responses, we calculated the pairwise Pearson correlation coefficients between the ancestor's responses and the effect sizes of different offspring generations. To assess the reproducibility of effect sizes between offspring generations and tests, we also computed pairwise Pearson correlation coefficients between offspring generations for each test and between the tests. These correlation coefficients were estimated for subsets of data, including significant effects in either generation only or excluding these significant effects. In addition, for Test I, we used subsets of data, including the effect sizes of only one genotype or only one treatment, to calculate correlation coefficients between generations. Visualization was conducted using the *ggplot* function in the R package "ggplot2" v3.4.2[62].

## Genome-wide gene expression analyses

To assess patterns of gene expression changes caused by ancestral treatments, we collected fresh leaf samples from plants in Test II and conducted transcriptomic analysis. This analysis included four genotypes (Abd-0, Ang-0, Tol-0, TRE-1), four environments (the control, drought, low cadmium, and high cadmium), four generations (F1-F4), each with three replicates, and a total of 144 samples. These genotypes and environments were carefully selected to include genotypes and treatments representing a high/low occurrence probability of transgenerational effects. For instance, Abd-0 and TRE-1 represent genotypes sensitive to environmental induction, whereas Ang-0 and Tol-0 represent genotypes resistant to environmental induction. We sampled two to three rosette leaves per plant, immediately froze the samples in liquid nitrogen, and stored them at −80 °C before RNA extraction. We extracted total RNA with TRIzol reagent (Invitrogen)[63]. The samples of total RNA were subjected to quality assessment with an Agilent 2100 Bioanalyzer. The transcriptome sequencing library was prepared with the NEBNext® UltraTM RNA Library Prep Kit for Illumina® (NEB, USA). The library was sequenced with the Illumina NovaSeq platform to produce 150 bp paired-end reads. Quantity assessment, library preparation and sequencing were conducted by

Novogene (Beijing, China). Sequencing yielded 19.5 to 29.4 million pairs of reads (average: 22.6) per sample (Supplementary Data 1).

To improve the quality of reads, we trimmed and filtered the reads with Trimmomatic v0.36 with the following parameters: "ILLUMINACLIP: TruSeq3-PE.fa:2:30:10" LEADING:20 TRAILING:20 SLIDINGWINDOW:4:20 MINLEN:50. Reads with improved quality were aligned to the TAIR10 reference genome using STAR v2.7.9[64] with the default parameters and a TAIR10 gff-converted gtf-format file (parameter '-sjdbGTFfile') (Supplementary Data 1). The conversion was conducted with gffread v.0.12.7[65]. We extracted the gene expression matrix using featureCounts v1.22.2[66], with the '-p' option to count the meta-features of genes using the TAIR10 gtf file. Genes were included in subsequent analysis if the sum CPM (counts per million) over all samples for the treatment and control per genotype was greater than one. The retained gene and the expression were subjected to cluster analysis of different samples with the *plotPCA* function in the "DESeq2" v1.36.0 package[67], and cluster analysis results were visualized using the R package "ggplot2". We subjected the retained genes and their expression to differential expression analysis with the R package DESeq2 v1.36.0 (these genes were also used for random sampling, please see below). For each genotype, the difference in gene expression was estimated by comparing the expression of a particular gene between the ancestral treatment and the control. The expression matrix was normalized by function *vst* in package "DESeq2". The difference in gene expression was estimated with function *DESeq*, and the differentially expressed genes (DEGs) were filtered by|log2FoldChange|>1.5 and adjusted *P* < 0.05. A gene was considered to be a heritable DEG if it was significantly differentially expressed for at least two generations. To explore the factors that explained the number of heritable DEGs, we regressed the log-transformed number per genotype and treatment against the arcsine-transformed occurrence frequency per genotype and treatment as well as the log-transformed number of transposons per genotype and extracted the correlation coefficient and *P* value from the regression.

To investigate the function of DEGs, we focused on genes that were differentially expressed in at least two offspring generations (heritable DEGs) and conducted KEGG and Gene Ontology (GO) enrichment analysis. The enrichment analysis was conducted with functions *enrichKEGG* and *enrichGO* in the package "clusterProfiler" v4.4.4[68]. Enrichment analysis was conducted separately for four genotypes and three environments, which resulted in 12 sets of enrichment results (Supplementary Data 2). For the KEGG results, we took the union of significantly enriched pathways of the 12 sets, clustered the pathways according to the maps and visualized the results. For the GO results, we removed duplicated terms for each set with function *simplify* in the package (Supplementary Data 3). Then, we filtered out terms with less than five genes enriched and selected the top 15 terms with the smallest *P* values for each set, took their union, and visualized the results. KEGG and GO visualization was conducted with the R package "pheatmap" v1.0.12 [https://github.com/raivokolde/pheatmap]. To facilitate visualization, we characterized the terms into functions of response to stimulus, biosynthetic and metabolic processes, cellular process, signalling, biological regulation, rhythmic process, and developmental process. Then, we extracted the mapped heritable DEGs in pathways and terms and summarized their overlapping patterns between genotypes and environments.

We found 2669 heritable DEGs shared between genotypes or environments, which are considered genes sensitive to environmental induction. We hypothesized that their sensitivity was related to the enrichment of transposons in their regulatory and gene body regions. To test this hypothesis, we randomly sampled 2669 genes from the genes subjected to the differential expression analysis and estimated the sum of the number of transposons per superfamily overlapping the 2 kb upstream and gene body regions. Through random sampling 10000 times, we obtained the null distribution of transposon

abundance upstream of genes for different superfamilies. Then, we compared the observed transposon abundance upstream of the 2669 heritable DEGs against this distribution. The *P* value was estimated as the proportion of randomly sampled values greater than the observed value, and significant *P* values indicate significant enrichment of transposons upstream of heritable DEGs compared with the null expectation. We adopted the transposon annotation of TAIR10 and performed the analysis using a customized script coded with R package "GenomicRanges" v.1.42.0[69] [available at https://github.com/Xiaohe-Lin/Heritable_variation/]. Notably, our study adopted Col-0 as the reference, and genotype-specific genome assembly is necessary for a comprehensive understanding of this nonrandom phenomenon.

To investigate whether the heritable phenotypic changes were caused by heritable gene expression changes, we focused on flowering time and identified 79 flowering time-related genes from the literature[14–16] (Supplementary Data 4). We visualized the expression patterns of a subset of genes that were DEGs for at least one generation using the R package "pheatmap" v1.0.12.

### Reporting summary
Further information on research design is available in the Nature Portfolio Reporting Summary linked to this article.

## Data availability
The phenotypic data of the ancestral generation, Test I and Test II are available on GitHub [https://github.com/Xiaohe-Lin/Heritable_variation]. Raw RNA-seq data are available from the NCBI SRA database with accession ID PRJNA997595. Source data are provided as a Source Data file. The geographical and climatic data sourced from WorldClim [https://worldclim.org], and information of DNA methylation and transposon abundance were obtained from published literature (Kawakatsu. et al. Cell, 2016, 166:492-505; Quadrana. et al. eLife, 2016, 5:e15716). The Genome, gene and Transposon annotations of *Arabidopsis thaliana* are from TAIR website [https://www.arabidopsis.org]. Source data are provided with this paper.

## Code availability
Codes for data processing are available on GitHub [https://github.com/Xiaohe-Lin/Heritable_variation] and Zenodo [https://zenodo.org/records/11020990]

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

## Acknowledgements

We thank Wenjing Hong, Gang Wang, and Rong Wang for their constructive suggestions. We thank Ming Zhou, Xianwen Ma, Jiayi Ding, and Xinwei Wang for their help in the growth room and Xiuxiu Wang and Haidong Qu for their technical assistance. This study was supported by the National Natural Science Foundation of China (32371558 and 32071485 to Y.-Y.Z.) and the Fundamental Research Funds for the Central Universities of China (20720210075 to Y.-Y.Z.).

## Author contributions

Y.-Y.Z., O.B., and V.L. conceived the idea of this project; Y.-Y.Z. J.J.Y. and X.L designed the experiments; Y-Y.Z. supervised and Q.Q.L. co-supervised the project; J.J.Y., Y.W., J.Y, and X.L. carried out the experiments; X.L, JJ.Y., and Y.W. analysed the data; X.L wrote the first draft of this manuscript, and all authors contributed to the revision of the manuscript.

## Competing interests

The authors declare no competing interests.
