## [Peer Review File · Nature Communications]

Environment-induced heritable variations are common in *Arabidopsis thaliana*Response to reviewers' Comments:

Reviewer #1:

“The authors reported transgenerational changes in several different *Arabidopsis* cultivars exposed to different stresses and found that most of the responses are genotype-dependent and correlate well with the transposon abundance in the genotypes. Overall, the work is convincing. I have some minor corrections/requests.

Response: Thank you for your positive comments and constructive suggestions. Following your suggestions, we have improved the method details and figures, and hope you will find the clarity of the manuscript further improved. Please refer to our point-to-point reply below, with comments in black, the reply in blue, and the revised texts highlighted.

Provide more detailed information (maybe I missed that) on the transposon occupancy in different cultivars, by type and genomic size. Show correlation for other transposons, for which the correlation was not as convincing.”

Response: Thank you for the suggestion. For the information on transposon occupancy, please refer to Supplementary Table 1. We revised the table structure and referred to it in the main text (Line 139) to make it more accessible.

In this table, the number of transposons per superfamily is given, which was extracted from Quadrana. *et al.* eLife, 2016, 5:e15716. This number was originally estimated from Illumina read coverage (Lines 484-488). To show the correlation for all transposon superfamilies, we revised the Supplementary Fig.3b (Test I) and Supplementary Fig. 7c (Test II). The corresponding correlation is also shown in Fig. 3b (Test I) and Supplementary Fig. 7b (Test II).

Supplementary Fig. 3b

Supplementary Fig. 7c

As the original paper did not provide size estimation, the transposon occupancy by genome size remained unknown. Recently, we reassembled two genotypes exhibiting the highest frequencies of induced variation, Abd-0 and TRE-1 (unpublished data), and found that the genome sizes are 135M for Abd-0 and 138M for TRE-1, much greater than reference genome Col-0 (119 M, TAIR10), and even greater than a recent T-to-T version of Col-0 (134 M, Hou *et al.*, Mol. Plant, 2022, 15, 8). Although not perfectly estimated in this manuscript, we hope that our attempt to correlate transposon occupancy with induced heritable change still can serve as an incentive to bridge the fields of transposon and transgenerational effect studies, and to connect old ideas with new observations, offering valuable insights for future studies (Please refer to our reply to your points below).

“..The frequency of occurrence of fruit number was...” – delete “number”; is it actually just seed amount? “the frequency of occurrence of fruit” sounds weird – sounds like some plants had no fruits.”

Response: Thanks for pointing out this issue. This is actually the number of siliques per plant. This phenotype is often adopted as a fitness proxy for *A. thaliana* because it is highly correlated with seed number, and counting the tiny and large quantity of *Arabidopsis* seeds is indeed challenging. To make this point clear, we have revised the sentence as follows: “The occurrence frequency estimated on the fitness proxy (=number of siliques per plant or fruit number) was the lowest among different phenotypes, marginally significantly lower than those of flowering time and plant height.” (Lines 117-120)

“the number of genes showing heritable expression changes can be predicted by the abundance of transposons in the genomes.” – I would use the word “correlate”

Response: Thanks. We revised accordingly as: “...the number of genes showing heritable expression changes correlates with the abundance of transposons in the genomes.” (Lines 221-223)

“It was not very clear why you grown it for 7 generations and tested after 4th. ”

Response: Thanks. This is because there were two generations to remove the potential effect from seed storage and one to conduct the environmental treatments. Thus, there are three generations, and in 4th generation, we tested whether the treatments induced changes in decendants. To clarify the design, we revised the main text as follows: “To address the prevalence and predictability of environment-induced heritable changes, we subjected 14 natural accessions (genotypes) of *Arabidopsis thaliana* to the Control and ten environmental treatments to establish the ancestral generation. Before this generation, we planted the seeds of these accessions in the control environment for two generations to remove potential influence due to seed collection or storage. Then, we cultivated the offspring of the ancestral generation in the control environment to assess whether those environmental treatments induced phenotypic changes in offspring and

whether the induced changes were heritable over four offspring generations (Test I, Fig. 1, Supplementary Fig. 1a, Supplementary Table 1).”(Lines 50-58)

“Please elaborate on pooling F1 – F4 seeds together for the Test II. Seeds from how many plants were pooled? By weight or by number?”

Response: Thank you for raising this point. Seeds from different plants or generations were not pooled, as we collected and reserved seeds for each plant and generation separately. We followed the single-seed descendant approach, that is to grow one individual descendent per plant per generation, to ensure there is no effect of selection. To clarify this point, we revised the methods as follows:

“Each treatment included six replicate plants per genotype, and the control included 12 plants per genotype, resulting in 1008 plants grown or lines established in the ancestral generation. Seeds were collected and reserved for each line, and in the offspring generation, we planted one descendant per line, following the single-seed descendant approach. In Test I, all 1008 lines were planted generation-by-generation in the control environment for four generations (F1-F4), and together 4032 plants were planted. For each offspring generation, seeds were collected and reserved for each line and generation separately.”(Lines 343-348)

Test II included a subset of ten genotypes (Abd-0, Col-0, Ang-0, Dja-1, Fei-0, Ler-0, Sapporo-0, Tol-0, TRE-1 and Tri-0), six different environments (the control, drought, low cadmium, high cadmium, low salinity and high salinity), and thus 360 lines (=10 genotypes * 6 environments * 6 replicates). For each line, we regenerated one plant from reserved seeds of the previous generation, and 1440 plants were grown to establish F1-F4.”(Lines 363-365)

“A better Discussion about the role of transposons in the evolution is needed; there are reports demonstrating that transposons are activated by stress and genomic changes

prevail in transposons. ”

Response: Thank you for your constructive suggestions. Accordingly, we have revised the discussion section to incorporate evidence demonstrating that transposons become active under stress, leading to notable genomic changes. For the role of transposon, we highlighted their potential dual function as both selectors and inducers of genetic variation. Additionally, we have integrated suggestions from Reviewer #2. Please refer to the revised discussion section below:

...Transposons are mobile elements that constitute a significant proportion of eukaryotic genomes²⁵. While these elements were once considered junk DNA, it became evident that they can contribute to the transcriptional binding region²⁶, methylation loci²¹, transcriptional isoform sites²⁷, and even coding sequences²⁸ to the host genomes. Barbara McClintock, who discovered transposons, proposed the idea of "genome shock," suggesting that stress can activate the movement of transposons, causing insertional mutations and structural variants²⁹. This viewpoint has been increasingly supported by evidence that stress increases transposition rate³⁰⁻³³, with the phenotypic effects of *de novo* transposon insertions including enhanced resistance of insects to pesticides³⁴⁻³⁶ and decreased sensitivity of plants to abiotic stress³⁶⁻³⁹. The stress-induced transposition may be mediated by Hsp70 chaperone, a response factor to various biotic and abiotic stressors in plants and animals^{40,41}, whose inducible expression could disturb the biogenesis of transposon-silencing piRNA⁴². These studies, along with our own, bridge and revive these previous ideas and explanations, suggesting that the environment is not only the selector but can also act as an inducer of genetic variation^{36,43}. (Lines 284-299)

... Some notable examples of rapid evolution based on *de novo* transposon insertion include the industrial melanism mutation of the peppered moth²³ and the rapid increase in resistance to DDT in *Drosophila melanogaster*²⁴. While many transposon insertions induced by environmental factors are deleterious, they enable the creation of alleles with significant, positively selected effects^{37,49,50}. If the environment can act as both a

selector and an inducer of heritable variation, then the evolutionary speed constraints imposed by standing genetic variations will be liberated^{43,48}. Therefore, exploring environmentally induced heritable variations and the molecular mechanisms related to transposon regulation is not only significant for fundamental research in genetics and evolution, but also crucial for assessing the threats of global changes and species responses.” (Lines 306-316)

Reviewer #2

“By such paper, the authors show that different types of environmental stress are capable to induce heritable phenotypic variation. Interestingly, it has also shown that such heritable changes are related to the abundance of transposons. Such data are interesting and significant together with other previous data, obtained on different organisms, that are changing the classical view on the environment as a major player only in selecting heritable changes. The very interesting point is that transposable elements are significant mediators in heritable changes induce by environmental stress. Such notion has been experimentally explored also in animal organisms. Among the several papers on such topic, I suggest to read some the following ones just to have a more complete view about the mechanisms involved on heritable changes produced by the environment through the activation of transposable elements:

Specchia V., L. Piacentini, P. Tritto, L. Fanti, R. D’Alessandro, G. Palumbo, S. Pimpinelli and MP. Bozzetti (2010). HSP90 prevents phenotypic variation by suppressing the mutagenic activity of transposons. *Nature* 463: 662-665.

Piacentini L., Fanti L., Specchia V., Bozzetti MP., Berloco M., Palumbo G., Pimpinelli S. (2014) Transposons environmental changes and heritable induced phenotypic variability. *Chromosoma*. 123(4):345-54.

Fanti L, Piacentini L, Cappucci U, Casale AM, Pimpinelli S. (2017) Canalization by Selection of de Novo Induced Mutations. *Genetics*. 206(4):1995-2006.

Cappucci U, Noro F, Casale AM, Fanti L, Berloco M, Alagia AA, Grassi L, Le Pera L, Piacentini L, Pimpinelli S. (2019) The Hsp70 chaperone is a major player in stress-induced transposable element activation. *Proc Natl Acad Sci U S A*. 116(36):17943-17950.

Pimpinelli S, Piacentini L. (2020) Environmental change and the evolution of genomes: Transposable elements as translators of phenotype plasticity into genotypic variability. *Functional Ecology* 34 (2), 426-441.

In conclusion, the present work is well done and show very significant results. In my opinion, it deserves to be published.”

Response: Thank you very much for your positive comments and for providing the references. These references have significantly expanded our knowledge of observations in various animals and plants, and the molecular mechanisms underlying stress-activating transposons. More importantly, these references have also summarized hallmark discoveries and bridges between those milestone ideas and concepts, offering thoughtful insights into this field. We have revised our discussion (please see below), and hope to put our findings into such a perspective.

...Transposons are mobile elements that constitute a significant proportion of eukaryotic genomes²⁵. While these elements were once considered junk DNA, it became evident that they can contribute to the transcriptional binding region²⁶, methylation loci²¹, transcriptional isoform sites²⁷, and even coding sequences²⁸ to the host genomes. Barbara McClintock, who discovered transposons, proposed the idea of "genome shock," suggesting that stress can activate the movement of transposons, causing insertional mutations and structural variants²⁹. This viewpoint has been increasingly supported by evidence that stress increases transposition rate³⁰⁻³³, with the phenotypic effects of *de novo* transposon insertions including enhanced resistance of insects to pesticides³⁴⁻³⁶ and decreased sensitivity of plants to abiotic stress³⁶⁻³⁹. The stress-induced transposition may be mediated by Hsp70 chaperone, a response factor to various biotic and abiotic stressors in plants and animals^{40,41}, whose inducible expression could disturb the biogenesis of transposon-silencing piRNA⁴². These studies, along with our own, bridge and revive these previous ideas and explanations, suggesting that the environment is not only the selector but can also act as an inducer of genetic variation^{36,43}. (Lines 284-299)

... Some notable examples of rapid evolution based on *de novo* transposon insertion include the industrial melanism mutation of the peppered moth²³ and the rapid increase in resistance to DDT in *Drosophila melanogaster*²⁴. While many transposon insertions induced by environmental factors are deleterious, they enable the creation of alleles with significant, positively selected effects^{37,49,50}. If the environment can act as both a selector and an inducer of heritable variation, then the evolutionary speed constraints imposed by standing genetic variations will be liberated^{43,48}. Therefore, exploring environmentally induced heritable variations and the molecular mechanisms related to transposon regulation is not only significant for fundamental research in genetics and evolution, but also crucial for assessing the threats of global changes and species responses.” (Lines 306-316)